# Effects of mesotrione on the control efficiency and chlorophyll fluorescence parameters of *Chenopodium album* under simulated rainfall conditions

**Mengmeng Sun[1]<sup>☉</sup>, Meijun Guo[2]<sup>☉</sup>, Shuai Guo[1], Yanfen Li[1], Shuqi Dong[1], Xie Song[1], Xiaoxin Shi[3], Xiangyang Yuan[1] \***

1 Laboratory of Crop Chemical Regulation and Chemical Weed Control, College of Agronomy, Shanxi Agricultural University, Taigu, Shanxi, China, 2 Department of Biological Science and Technology, Jinzhong University, Jinzhong, Shanxi, China, 3 Taigu County Meteorological Bureau, Shanxi, China

☉ These authors contributed equally to this work.

\* yuanxiangyang200@163.com

**Data Availability Statement:** All relevant data are within the paper and its Supporting Information files.

## Abstract

This experiment was conducted to study the effects of mesotrione on the control efficiency and chlorophyll fluorescence parameters of *Chenopodium album*. Simulating three rainfall intensities of 2 mm/h (light rain), 6 mm/h (moderate rain) and 10 mm/h (heavy rain) at different interval times (0.5 h, 1 h, 2 h, 4 h) to analyze variable regulation of the control effect, the photosynthetic pigment content and chlorophyll fluorescence parameters of *C. album* after spraying mesotrione. With the extension of rainfall time interval, the inhibition rate of plant height, plant control effect and fresh weight control effect of *C. album* were gradually increased, the inhibition effect of rainfall on the efficacy was gradually decreased, at the same time, the contents of chlorophyll a, chlorophyll b, carotenoids, the maximum photochemical quantum efficiency (Fv/Fm), the actual photochemical quantum yield (Y (II)) and quantum yield (Y (NO)) production of regular energy consumption of *C. album* were also increased, while the nonregulatory energy decreased gradually. The results showed that the contents of chlorophyll a and chlorophyll b in leaves of *C. album* increased significantly by 35.63% and 35.38% compared with the control under the condition of simulating 6 mm/h in interval 1 hour. The study suggested that simulating 10 mm/h rainfall intensity had the greatest effect on *C. album*, the photosynthetic pigment content, Fv/Fm and Y (II) of leaves were significantly higher than those in the control groups under 0.5 h, 1 h and 2 h interval treatments. The carotenoid content was the lowest and Y (NO) was the largest under the 4 h interval treatment. As is displayed that rainfall reduced the weed control effect in the aspect of controlling *C. album* on mesotrione, which is partly contributed to increase photosynthetic pigment content and enhance the PS II photochemical activity. In conclusion, the rain intensity of ≤2 mm/h did not affect the control effect of mesotrione on *C. album*. At 6 mm/h within 1 h after treatment, the control effect of fresh weight was significantly reduced by more than 7.14%, and at 10 mm/h within 2 h, the control effect was significantly reduced by more than 14.78%.

**Funding:** This work was supported by the system of National Modern Agriculture Technology (CARS-06-13.5-A2 and CARS-06-14.5-A28, awarded to XY), the Program for the Top Young Innovative Talents of Shanxi Agricultural University (TYIT201406 awarded to XY), the Key research and development plan of Shanxi Province (201903D221030 awarded to XS) and Research Program Sponsored by State Key Laboratory of Sustainable Dryland Agriculture (in preparation), Shanxi Agricultural University (202003-5 awarded to XY). The funders had no role in study design, data collection and analysis, decision to publish, or preparation of the manuscript.

**Competing interests:** The authors have declared that no competing interests exist.

# 1 Introduction

Weed would reduce the yield and quality of the crops, hinder mechanized sowing and harvest and then slow down the process of agricultural modernization [1,2]. *Chenopodium album*. L is an erect annual herbaceous plant, that grows between 0.2 and 2 meters in height, striped green, red or purple stems. Leaves are rhomboid, deltoid to lanceolate, upper entire, lower toothed or irregularly lobed. The leaves are wax-coated, sometimes has mealy and unregular sawtooth on the edge of them, with a whitish coat on the underside. The length of its petiole is close to blade or half of the blade [3]. *C. album* is one of the common farmland malignant weed in northern China. It has strong environmental adaptability and a large root system, even grow on the Tibetan Plateau and in the lowlands of southwest China, so the *C. album* is recorded as a potherb in both plateaus and low lands [4]. It has obvious advantages when competing with crops for water and fertilizer, every single plant could yield 3000–20000 seeds (Data from CABI), which led to a sharp decline in crop yield [5,6]. Sugarbeet, barley, mustard, gram, maize, foxtail millet and so many crops were defeated by *C. album*, it has been reported to reduce the soybean (*Glycine max*) yield by 61%, in the wheat (*Triticum aestivum*) field, the yield loss approximately 50–60%, which also was associated with interference of this weed [7–9]. When the density increases to 20 plants per $m^2$, the corn would not ear because of the shade effect of *C. album* [10].

Mesotrione is a herbicide of broad-spectrum, selective stem and leaf before bud and after seedling mainly used to effectively control broad-leaved weeds and some gramineous weeds by inhibiting the catalytic factors of plant photosynthetic process. Mesotrione is widely used in weed control of maize and winter wheat due to its high activity, low residue, strong compatibility and safety to the environment and subsequent crops [11,12]. Summer is the key period for crop growth and weeding. However, local microclimate and severe convective weather occur frequently, and rainfall interfere the herbicide efficacy due to the scouring every year [13]. Therefore, it is attached importance to study the influence of rainfall on herbicides. There are great differences in the absorption rate and rain resistance among different stem and leaf treatment agents, so the interval time of affecting the efficacy is different, but most herbicides will not affect the weed when rainfall after 4–6 h [14]. Wu *et al*. [15] found that rainfall after the application had a significant inhibitory effect on the efficacy of nicosulfuron methyl; paraquat could effectively control *Alopecurus aequalis* after 0.5 h and 1 h rainfall [16]; there was no significant difference in bensulfuron efficacy between without rainfall and after 2 h rainfall [17]; Wang *et al*. [18] showed that the effect of imazethapyr was no longer affected under the 9 mm/h rainfall intensity. When the interval time of rainfall is 0.5 h, the effect of rainfall on barnyardgrass was significantly reduced, it's down to 19.42% [19]. At present, it is known that mesotrione has effects on the growth of weeds and crops, such as reducing the plant height of maize, making *Amaranthus retroflexus* heart leaves be yellow, reducing the photosynthetic pigment content, reducing the maximum photochemical quantum efficiency (Fv/Fm) and the actual photochemical quantum yield Y (II) [20–22]. The inhibition rate of plant height, plant control effect and fresh weight control effect can well reflect the control effect of herbicides on weeds; photosynthetic pigment content reflects the green extent of plants; chlorophyll fluorescence parameters as a probe of photosynthetic capacity of leaves can reflect the status of weeds under herbicide stress [23]. Most of the studies on mesotrione focused on its control effect and crop safety. There were no reports on the control effect and chlorophyll fluorescence parameters of mesotrione on broad-leaved weed *C. album* under rainfall conditions. In this experiment, artificial rainfall was used to simulate rainfall, and *C. album* is a common malignant weed in the field, was used to simulate different rain intensities at different intervals after spraying mesotrione.

**Table 1. Meteorological data of the experimental sites during May-September in 2019.**

| Month | Precipitation (mm) | The max rainfall intensity (mm/h) | Temperature (°C) | | | ≥20°C accumulated temperature (°C) | Sunshine hours (h) |
|---|---|---|---|---|---|---|---|
| | | | Average | Min | Max | | |
| 5 | 1 | 0.8 | 19.9 | 35.3 | 2.6 | 235.1 | 258.9 |
| 6 | 55.7 | 9.3 | 24.5 | 35.6 | 12.4 | 715 | 213.4 |
| 7 | 65.1 | 13.4 | 25.4 | 37.1 | 10.8 | 788.4 | 243.1 |
| 8 | 52 | 5.2 | 22.9 | 34.5 | 9.7 | 614.2 | 219.6 |
| 9 | 78.8 | 6 | 18.4 | 34.1 | 6.6 | 195.5 | 223.9 |

In order to assess the possibility of mesotrione application and understand the related mechanism, we investigated the weed control, agronomic traits, photosynthetic pigment content and the chlorophyll fluorescence parameters of *C. album* to mesotrione in this current study. The influence of rain intensity and interval time on the efficacy was determined, which provide a theoretical basis for the use of mesotrione.

## 2 Materials and methods

### 2.1 Experimental site

Field experiments were conducted at farming station of Shanxi Agricultural University during 2019, Jinzhong, China (37.43°N, 112.61°E). The area is located in the northeast of Jinzhong region and belongs to a warm temperate continental monsoon climatic with an altitude of approximately 800 meters, the annual average temperature was 9.9°C, and the frost-free period was about 159 days. Annual precipitation of the province was 400–650 mm, but the seasonal distribution was uneven, with more than 60% of precipitation concentrated in June to August [24]. Tables 1 and 2 present the meteorological conditions and soil characteristics of experimental site.

### 2.2 Experimental design

Four rainfall intervals were set: 0.5 h (T1), 1 h (T2), 2 h (T3), 4 h (T4), and no rainfall after mesotrione application was taken as control (CK). The randomized block design was conducted by the field experiment with three replicates in 4 $m^2$ (2 m × 2 m). Mesotrione (10%, OD) was provided by Shandong Shengbanglvye Chemical Limited Company, the recommended effective dose given by the manufacturer was 150 g ai $ha^{-1}$. The application was performed with a laboratory pot sprayer to deliver 450 L $ha^{-1}$.

### 2.3 Rainfall designs

According to the standard of China Meteorological Administration(CMA): the hourly rainfall is 0.1~2.5 mm for light rain, 2.6~8 mm for moderate rain, and 8.1~15.9 mm for heavy rain. Simulating light rain 2 mm/h, moderate rain 6 mm/h and heavy rain 10 mm/h by power operated sprayers (Lufeng, 3 WBD-20 C, 20 L). According to the calculation method that 1 L of

**Table 2. The basic physical and chemical properties of soil.**

| Depth of soil layer | pH | Available potassium (mg/kg) | Available phosphorus (mg/kg) | Available nitrogen (mg/kg) | organic matter (g/kg) |
|---|---|---|---|---|---|
| 0–5 cm | 8.21 | 484.21 | 43.70 | 64.20 | 8.52 |
| 5–10 cm | 8.14 | 471.93 | 26.79 | 42.80 | 8.52 |
| 10–15 cm | 8.17 | 319.60 | 21.28 | 49.93 | 7.99 |

rainfall per square meter is 1 mm [25], artificial spraying was carried out by simulating natural rainfall, each experiment plot featured a dimension of 2 m ×2 m and the height of the spray is 0.5 meters. By adjusting the knob on the edge of the sprayer, 4 L, 12 L, and 20 L of water are sprayed out within half an hour to obtain three rain intensities of 2 mm/h, 6 mm/h and 10 mm/h. Before the experiment, several tests were carried out with the meteorological rain gauge. Firstly, we screened out one type of shower that sprayed the smallest water and most homogeneous when spraying, then the positions of different rain intensities on the sprayer knob were accurately marked. The process was repeated until the desired rainfall amounts and intensities were achieved.

To prevent drift effects in field conditions, we put many baffles around the rainfall simulation areas of the neighborhood. The simulated rainfall is carried out under calm or breezy weather to reduce the error, and the test rain intensity is determined again every time the simulated rainfall is performed, the error of these results is within 0.2 mm/h.

## 2.4 Data collections

**2.4.1 Determination of weed control effect.** To calculate the reductions in weed number and biomass, the shoots of all *C. album* plants were cut from three 0.25 m$^2$ (0.5 m × 0.5 m) quadrats in each plot (2 m × 2 m) at 15 and 30 days after treatment (DAT) [26]. When mesotrione in soil was analyzed, *C. album* were collected carefully by taking out from the soil in the open field. The plant was weighed after washing, the biomass of each weed sample were measured. The efficacy and reduction was calculated by this equation:

$$Efficacy\ (\%) = [(C - B)/C] \times 100 \tag{1}$$

where C is the aboveground height of the nontreated control plot and B is the height of an treated plot.

$$Reduction\ (\%) = [(SCP - TP)/SCP] \times 100 \tag{2}$$

Where TP is the weed plant number or biomass in the treated plots and SCP is the sum of the plant number or biomass in the weedy controls.

**2.4.2 Determination of photosynthetic pigment content.** Photosynthetic data was collected after 3, 6 and 9 d, weight 0.05 g of mature and intact leaves with the same growth in the middle and upper part of the plant, cut them into pieces, and put them into a stoppered test tube containing 5 ml of 96% ethanol, and soak in a dark place at room temperature for 48 hours until the leaves are milky white and oscillate for several hours during the extraction process. The absorbance of the extract is measured at wavelengths of 470 nm, 649 nm and 665 nm, and the content of chlorophyll a, chlorophyll b, and carotenoid are calculated [27]. The calculation formulas are in the following:

$$Ca = 13.95A_{665} - 6.88A_{649} \tag{3}$$

$$Cb = 24.96A_{649} - 7.32A_{665} \tag{4}$$

$$Cx.c = (1000A_{470} - 2.05\ Ca - 114.8\ Cb)/245 \tag{5}$$

**2.4.3 Determination of chlorophyll fluorescence parameters.** The chlorophyll fluorescence parameters were tested at 12 h, 24 h, 2 d and 4 d after application respectively. The middle and upper parts of the plant were selected to be fully stretched and mature leaves. The MINI-PAM-II portable pulse modulated chlorophyll fluorometer (Walz, Germany), first use

the leaf clip of the fluorometer to clamp the *C. album* for 30 min. Actual photochemical efficiency Y (II), maximum photochemical quantum yield Fv/Fm of leaves, regulated energy dissipation quantum yield Y (NPQ) and non-regulated energy dissipation quantum yield Y (NO) were examined [28].

### 2.5 Statistical analyses

DPS 6.5 Analysis System was used to conduct all statistical analyses. Data were presented as mean ± standard error of the mean. Duncan's test was used to analyze significant differences among treatments in growth parameters, chlorophyll content and fluorescence parameters.

## 3 Results

### 3.1 The visual control effect

The reaction symptoms of *C. album* to the mesotrione become more serious with the time going of application. With the extension of rainfall interval, the growth of *C. album* was severely inhibited, the tip of leaves was more serious, and the heart leave was even completely withered and shed. The survey of weed injury level (Table 3) showed that rainfall reduced the herbicide injury of weed by 1~2 levels at 15 DAT. The weed damage decreased to grade 5 when the rain intensity was 10 mm/h and the interval was 0.5 h at 30 DAT, but other intervals is grade 6.

### 3.2 The effect of simulated rainfall on mesotrione

After spraying mesotrione, *C. album* exhibits disease like chlorosis and albino of stems and leaves. With applying the herbicide, the rainfall causes the plant height inhibition rate to decrease, and enhanced gradually increases with the extension of the interval. It is represented in (Table 4) that when the rain intensity is 2 mm/h, at 15 DAT, compared with CK, T1,T2 and T3 cause a significant reductions in the inhibition rate of plant height (11.80%, 10.87%, 9.70%, respectively). At 30 DAT, there were no significant difference between treatments and CK. When the rain intensity was 6 mm/h, after 30 days, the inhibition rate of the treatment with

**Table 3. Weed injury level at 15, 30 d after application of mesotrione and simulated rainfall.**

| Rainfall intensity | Interval time | Level of weed injury | |
|---|---|---|---|
| | | **15 d** | **30 d** |
| 2 mm·h⁻¹ | CK | 6 | 6 |
| | T1 | 5 | 6 |
| | T2 | 5 | 6 |
| | T3 | 6 | 6 |
| | T4 | 6 | 6 |
| 6 mm·h⁻¹ | CK | 6 | 6 |
| | T1 | 4 | 6 |
| | T2 | 5 | 6 |
| | T3 | 5 | 6 |
| | T4 | 5 | 6 |
| 10 mm·h⁻¹ | CK | 6 | 6 |
| | T1 | 4 | 5 |
| | T2 | 4 | 6 |
| | T3 | 5 | 6 |
| | T4 | 6 | 6 |

**Table 4. Effect of simulated rainfall on the weed control of mesotrione.**

| Rainfall Intensity | Interval time | Plant height inhibition rate/% | | | | Plant control efficacy/% | | | | Fresh weight control efficacy/% | | | |
|---|---|---|---|---|---|---|---|---|---|---|---|---|---|
| | | 15 d | Reduced efficiency | 30 d | Reduced efficiency | 15 d | Reduced efficiency | 30 d | Reduced efficiency | 15 d | Reduced efficiency | 30 d | Reduced efficiency |
| 2 mm·h⁻¹ | CK | 65.29 ±1.81a | —— | 84.29 ±1.14a | —— | 83.09 ±1.78a | —— | 96.42 ±3.39a | —— | 81.91 ±2.71a | —— | 96.67 ±3.01a | —— |
| | 0.5 h | 53.49 ±2.10b | 18.07 | 80.79 ±2.11a | 4.15 | 68.03 ±4.22b | 18.12 | 91.20 ±5.12a | 5.41 | 68.60 ±2.74b | 16.25 | 93.89 ±2.83a | 2.88 |
| | 1 h | 54.42 ±1.00b | 16.65 | 81.85 ±3.06a | 2.89 | 71.26 ±3.73b | 14.24 | 93.68 ±2.80a | 2.54 | 70.53 ±5.82b | 13.89 | 94.39 ±2.85a | 2.36 |
| | 2 h | 55.59 ±1.95b | 14.86 | 82.37 ±2.24a | 2.28 | 77.92 ±5.58ab | 6.22 | 95.32 ±1.37a | 1.14 | 76.45 ±3.67ab | 6.67 | 95.28 ±3.54a | 1.44 |
| | 4 h | 62.12 ±2.68a | 4.86 | 83.94 ±1.89a | 0.42 | 81.84 ±3.38ab | 1.50 | 96.08 ±7.82a | 0.35 | 80.26 ±4.78a | 2.01 | 95.74 ±2.53a | 0.96 |
| 6 mm·h⁻¹ | CK | 65.29 ±1.81a | —— | 84.29 ±1.14a | —— | 83.09 ±1.78a | —— | 96.42 ±3.39a | —— | 81.91 ±2.71a | —— | 96.67 ±3.01a | —— |
| | 0.5 h | 41.19 ±1.42c | 36.91 | 78.72 ±2.67b | 6.61 | 43.00 ±4.39c | 48.25 | 76.98 ±4.01c | 20.16 | 51.16 ±4.82c | 37.54 | 80.33 ±3.20c | 16.90 |
| | 1 h | 43.27 ±2.06c | 33.73 | 81.02 ±1.84ab | 3.88 | 56.69 ±2.46b | 31.77 | 85.21 ±7.30b | 11.60 | 54.92 ±4.39c | 32.95 | 89.77 ±6.00b | 7.14 |
| | 2 h | 50.71 ±1.72b | 22.33 | 82.14 ±2.46a | 2.55 | 63.30 ±3.94b | 23.82 | 89.50 ±5.51b | 7.18 | 64.42 ±3.43b | 21.35 | 94.28 ±3.68a | 2.47 |
| | 4 h | 54.83 ±1.16b | 16.02 | 83.44 ±1.81a | 1.01 | 65.73 ±2.37b | 20.89 | 90.41 ±4.72b | 6.23 | 68.74 ±4.01b | 16.07 | 96.28 ±2.15a | 0.40 |
| 10 mm·h⁻¹ | CK | 65.29 ±1.81a | —— | 84.29 ±1.14a | —— | 83.09 ±1.78a | —— | 96.42 ±3.39a | —— | 81.91 ±2.71a | —— | 96.67 ±3.01a | —— |
| | 0.5 h | 40.32 ±2.38c | 38.24 | 76.76 ±1.98c | 8.93 | 39.20 ±3.63c | 52.82 | 68.96 ±3.63d | 28.48 | 43.09 ±2.45c | 47.39 | 77.37 ±2.12c | 19.96 |
| | 1 h | 42.86 ±1.61bc | 34.35 | 79.38 ±2.93bc | 5.83 | 42.96 ±1.92c | 48.30 | 78.66 ±2.51c | 18.42 | 55.12 ±5.23b | 32.70 | 88.20 ±4.31b | 8.76 |
| | 2 h | 47.16 ±1.05bc | 27.77 | 82.12 ±1.30ab | 2.57 | 55.90 ±2.61bc | 32.72 | 82.17 ±1.53c | 14.78 | 57.16 ±4.23b | 30.22 | 89.87 ±3.39b | 7.03 |
| | 4 h | 53.94 ±0.99b | 17.38 | 82.36 ±1.28ab | 2.29 | 65.46 ±1.17b | 21.22 | 87.68 ±3.43b | 9.06 | 62.08 ±3.62b | 24.21 | 94.09 ±4.09a | 2.67 |

Data are shown as mean ±SE with the different lowercase letters in the same column represent significant differences (P<0.05).

only T1 interval significantly decreased by 5.57% compared with CK, while the other treatments had no significant difference compared with CK. In addition to, with the increase of rainfall intensity, there was significantly reduced by 7.53% and 4.91% at T1 and T2 in the inhibition rate of plant height in 10 mm/h, the reduced efficiency were 8.93% and 5.83%, respectively.

The control effect and fresh weight control effect of the plants are belong to the weed control effects, which is increased significantly at 30 DAT compared with 15 DAT. At 15 DAT, the weed control effects of only T1 and T2 were significantly lower than that of CK by 15.06%, 11.83%, 13.31%, and 11.38% in 2 mm/h, the reduced efficiency was more than 13 percent; the rain intensity was at 6 mm/h and 10 mm/h, the weed control effects were significantly lower than CK. At 30 DAT, when the rain intensity was 2 mm/h, the weed control effects were still more than 90% at all treatments. However, the fresh weight control effect of T1 and T2 treatments is significantly lower than CK by 16.34% and 6.90% in 6 mm/h, the reduced efficiency were 16.90% and 7.14%, respectively; compared with CK, T1, T2 and T3 were reduced in the fresh weight control (19.30%, 8.43%, 6.80%, respectively) when the rain intensity is 10 mm/h, and the reduced efficiency was above 7%.

### 3.3 Effects of simulated rainfall on photosynthetic pigment content of *C. album* leaves after spraying mesotrione

**3.3.1 The effect of simulated rainfall on the carotenoid content of *C. album* after spraying mesotrione.** The data in Fig 1 showed that the corresponding rainfall intensity was 2 mm/h, compared with CK at 3 DAT, T1 and T2 treatments show a significant increase, 24.73% and 17.2%. Compared with all interval treatments, CK at 6 DAT did not see a big difference, which could indicate that the rainfall in this period is not capable of preventing the albino process of *C. album*.

When the rainfall intensity reached 6 mm/h, it could find that the corresponding carotenoid content of T1 and T2 treatments increased at 3 and 6 DAT, and the ranges were 21.85%,

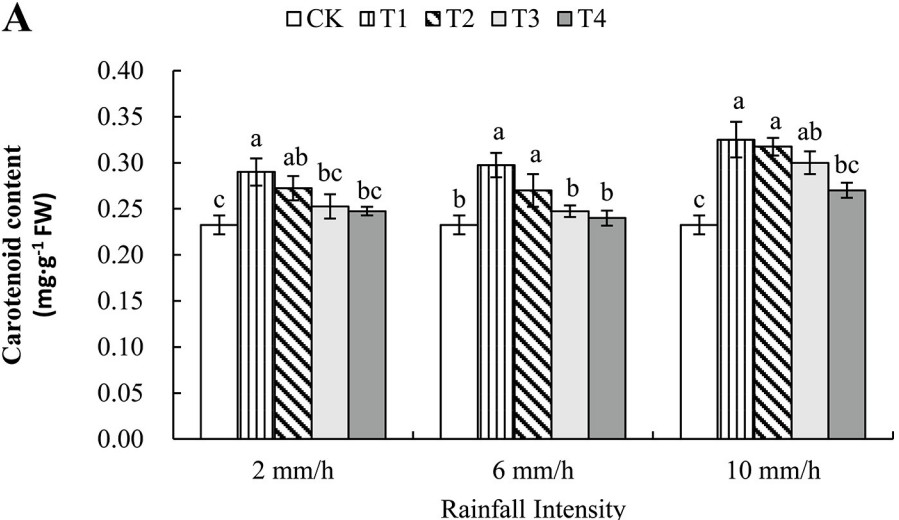

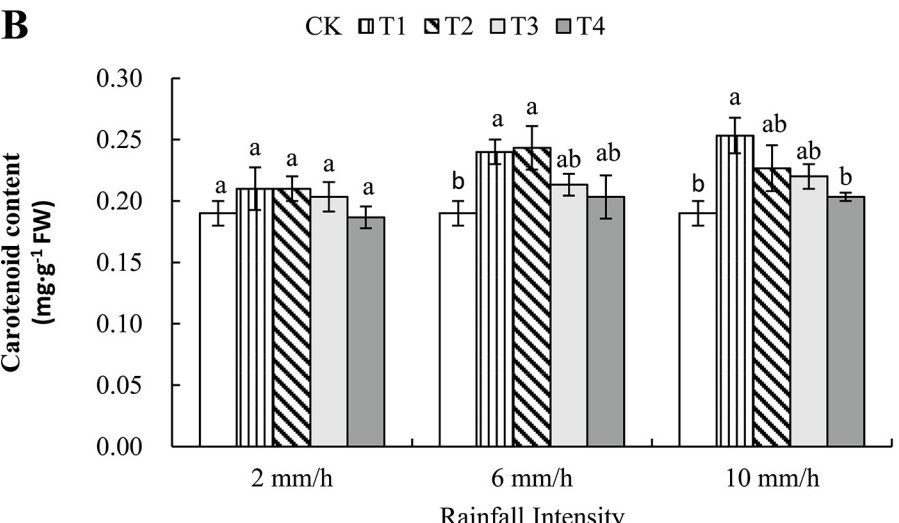

**Fig 1.** Effect of simulated rainfall on carotenoid content in leaves of *C. album* treated by mesotrione after 3 (A) and 6 (B) DAT. X-axis represents different rainfall intensity; y-axis represents the carotenoid content of leaves. CK means that there was no rain after the application of mesotrione, T1, T2, T3, and T4 represent different rainfall interval times (0.5 h, 1 h, 2 h, and 4 h). Effect of simulated rainfall on carotenoid content in leaves of *C. album* treated by mesotrione after 3 DAT (A) and 6 DAT (B). Lowercase letters in the same column indicate significant differences at the 0.05 level.

13.89%, 20.00% and 21.91%, respectively. Except that the T4 interval treatment was a bit different, the carotenoid content involved in other treatments looked higher than the content contained in CK. And the growth rates are 39.78%, 36.56%, 29.03% and 33.32%, 19.32%, 15.79%.

### 3.3.2 The effect of simulated rainfall on the chlorophyll content of *C. album* after spraying mesotrione

Like the carotenoid content, the content of chlorophyll a and chlorophyll b decreased gradually and showed a decreasing trend with the extension of rainfall interval. As shown in Table 5, when the rain intensity was 2 mm/h, the chlorophyll a and chlorophyll b contents of all interval treatments were not significantly different than CK at 6 DAT. At the same time, chlorophyll a +b content under T1 treatment has an increase of 27.86% compared with CK, the difference was not very significant between the other treatments and CK.

Under the situation of 3 DAT, the rainfall intensity reached 6 mm/h. In this case, in addition to the different results produced by T4 processing, the chlorophyll a content corresponding to other treatments has a more significant increment compared to CK. In addition, when the T1 and T2 intervals were used for treatment, the chlorophyll b content would increase significantly, with an increase of 46.41% and 35.38%. On the other hand, all treatments at certain intervals show that the contents of chlorophyll a and chlorophyll b were much higher than that situation of CK at 6 DAT. It could be claimed that compared with carotenoids, 6 mm/h rainfall had a greater impact on chlorophyll a and b. Although the interval is only 4 hours, it has a higher content than CK. When the rainfall intensity under 3 DAT reached 10 mm/h, the chlorophyll a content corresponding to all treatment intervals was at a high level, exceeding the chlorophyll content of CK. In the meantime, there was no significant difference between T1, T2 and T3 treatment.

## 3.4 Effects of simulated rainfall on chlorophyll fluorescence parameters of *C. album* after spraying mesotrione

**3.4.1 The effect of 2 mm/h rain intensities on chlorophyll fluorescence parameters after spraying mesotrione.** These two types of indicators Fv/Fm and Y (II) in the leaves showed a

**Table 5. Effect of simulated rainfall on photosynthetic pigment content in leaves of *C. album* treated by mesotrione.**

| Rainfall Intensity | Interval time | Chlorophyll a content (mg·g⁻¹ FW) | | Chlorophyll b content (mg·g⁻¹ FW) | | Chlorophyll a+b content (mg·g⁻¹ FW) | |
|---|---|---|---|---|---|---|---|
| | | 3 d | 6 d | 3 d | 6 d | 3 d | 6 d |
| 2 mm·h⁻¹ | CK | 0.92±0.058b | 0.79±0.101a | 0.27±0.020c | 0.26±0.027a | 1.19±0.040c | 1.05±0.075b |
| | T1 | 1.39±0.021a | 1.00±0.078a | 0.46±0.012a | 0.35±0.062a | 1.85±0.012a | 1.35±0.019a |
| | T2 | 1.06±0.098b | 0.90±0.049a | 0.38±0.012b | 0.31±0.030a | 1.44±0.087b | 1.21±0.020ab |
| | T3 | 1.09±0.074b | 0.82±0.074a | 0.34±0.023b | 0.32±0.027a | 1.43±0.054b | 1.14±0.048b |
| | T4 | 0.95±0.040b | 0.79±0.045a | 0.35±0.0208b | 0.28±0.025a | 1.30±0.030bc | 1.07±0.069b |
| 6 mm·h⁻¹ | CK | 0.92±0.058b | 0.79±0.101b | 0.27±0.020c | 0.26±0.027b | 1.19±0.04c | 1.05±0.075b |
| | T1 | 1.29±0.075a | 1.18±0.078a | 0.51±0.021a | 0.47±0.009a | 1.80±0.01a | 1.65±0.086a |
| | T2 | 1.24±0.031a | 1.14±0.067a | 0.37±0.035b | 0.46±0.012a | 1.57±0.03b | 1.60±0.055a |
| | T3 | 1.20±0.038a | 1.14±0.127a | 0.35±0.020bc | 0.42±0.018a | 1.59±0.06b | 1.56±0.111a |
| | T4 | 0.98±0.050b | 0.97±0.051ab | 0.35±0.037bc | 0.31±0.055b | 1.34±0.01c | 1.28±0.015b |
| 10 mm·h⁻¹ | CK | 0.92±0.058c | 0.79±0.101b | 0.27±0.020c | 0.26±0.027c | 1.19±0.040c | 1.05±0.075b |
| | T1 | 1.33±0.064a | 1.21±0.079a | 0.53±0.036a | 0.51±0.031a | 1.86±0.099a | 1.72±0.071a |
| | T2 | 1.34±0.064ab | 1.19±0.041a | 0.40±0.003b | 0.49±0.021a | 1.75±0.067a | 1.68±0.056a |
| | T3 | 1.14±0.020ab | 1.16±0.113a | 0.38±0.022b | 0.44±0.026b | 1.51±0.041b | 1.57±0.139a |
| | T4 | 1.18±0.083b | 1.15±0.050a | 0.36±0.035b | 0.37±0.019b | 1.54±0.050b | 1.53±0.037a |

Data are shown as mean ±SE with the different lowercase letters in the same column represent significant differences (P<0.05).

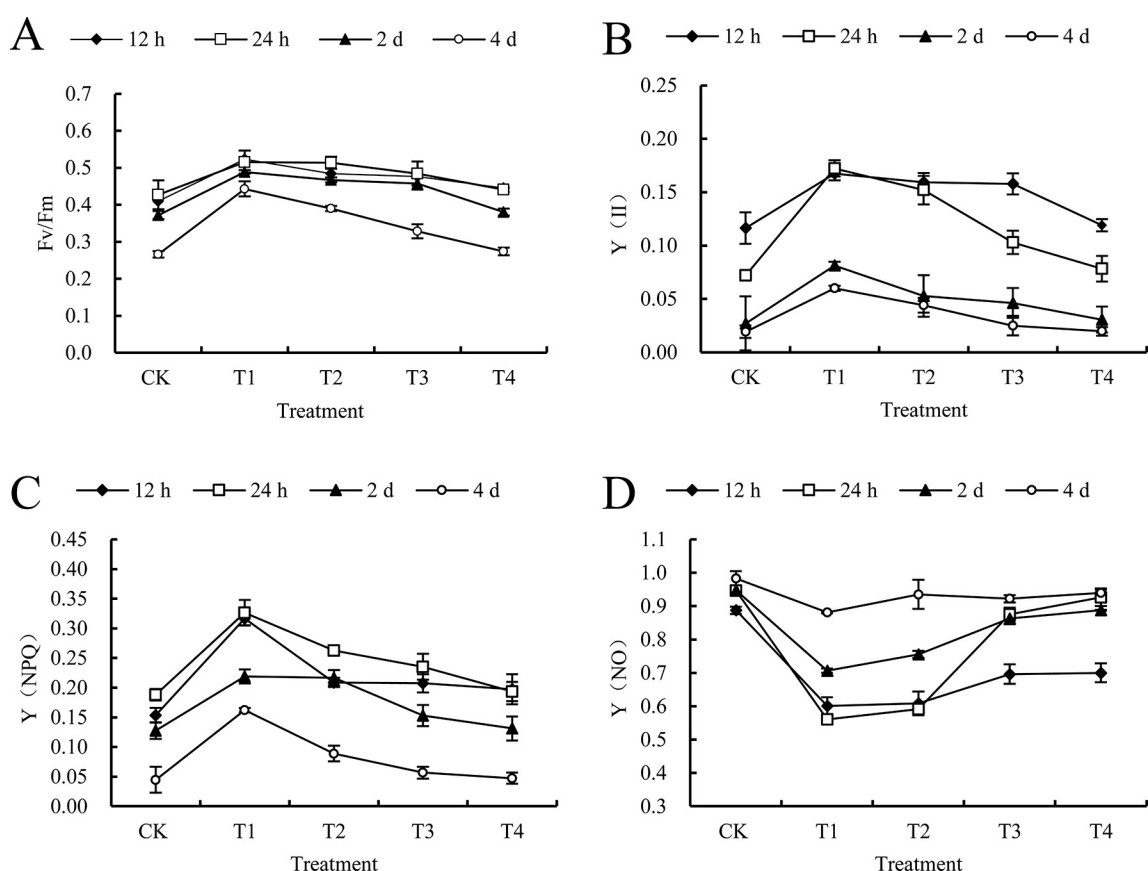

**Fig 2. Effect of 2 mm/h rain intensity on chlorophyll fluorescence parameters of *C. album* treated by mesotrione.** CK and T1, T2, T3, T4 on the x-axis represent no rain and different rainfall interval times (0.5 h, 1 h, 2 h, and 4 h) respectively; y-axis represents chlorophyll fluorescence parameters. Diamonds, squares, triangles and circles represent different times after treatment (12 h, 24 h, 2 d and 4 d, respectively). A) the maximum photochemical quantum efficiency (Fv/Fm); B) the actual photochemical quantum yield Y(II); C) the regulated energy dissipation quantum yield Y(NPQ); D) the non-regulated energy dissipation quantum yield Y(NO) of *C. album* treated by mesotrione. Lowercase letters in the same column indicate significant differences at the 0.05 level.

certain decline after using mesotrione. At the same time, all processing intervals were showing an upward trend. However, if the time between rains continued to extend, there was a possibility that there might be an overall decline between the two (Fig 2). The max value is T1 after the mesotrione, and the minimum will be T4. After taking relevant treatments, there is no significant difference in Fv/Fm indicators of other treatments compared with CK. But this does not include T1 treatment, T1 treatment caused a significant increase in Fv/Fm, with an amplitude of 27.65%.

In addition, it can be seen that there is no big difference between T4 interval treatment Y (II) and CK, and the remaining treatment Y (II) has a greatly increasment, whose amplitude are 43.85%, 36.97% and 35.60%, respectively. On the 4[th] day, if T1 treatment is not considered, the Fv/Fm of other treatments all show a certain increase. Compared with CK, the corresponding incidence rates are 66.54%, 46.62% and 23.42%, respectively; In the figure, the Y (II) treatment at T1 and T2 interval both increased to a certain extent, which increased by 210.88% and 127.98% respectively (Fig 2A and 2B). In addition, after the use of mesotrione, the entire graph trend showed that Y (NPQ) would decrease with the extension of the rain interval, but the value of Y (NO) would continue to increase. If T1 was not considered at 4 DAT, the Y (NPQ) of T2 interval treatment would increase significantly, and there might not be much difference

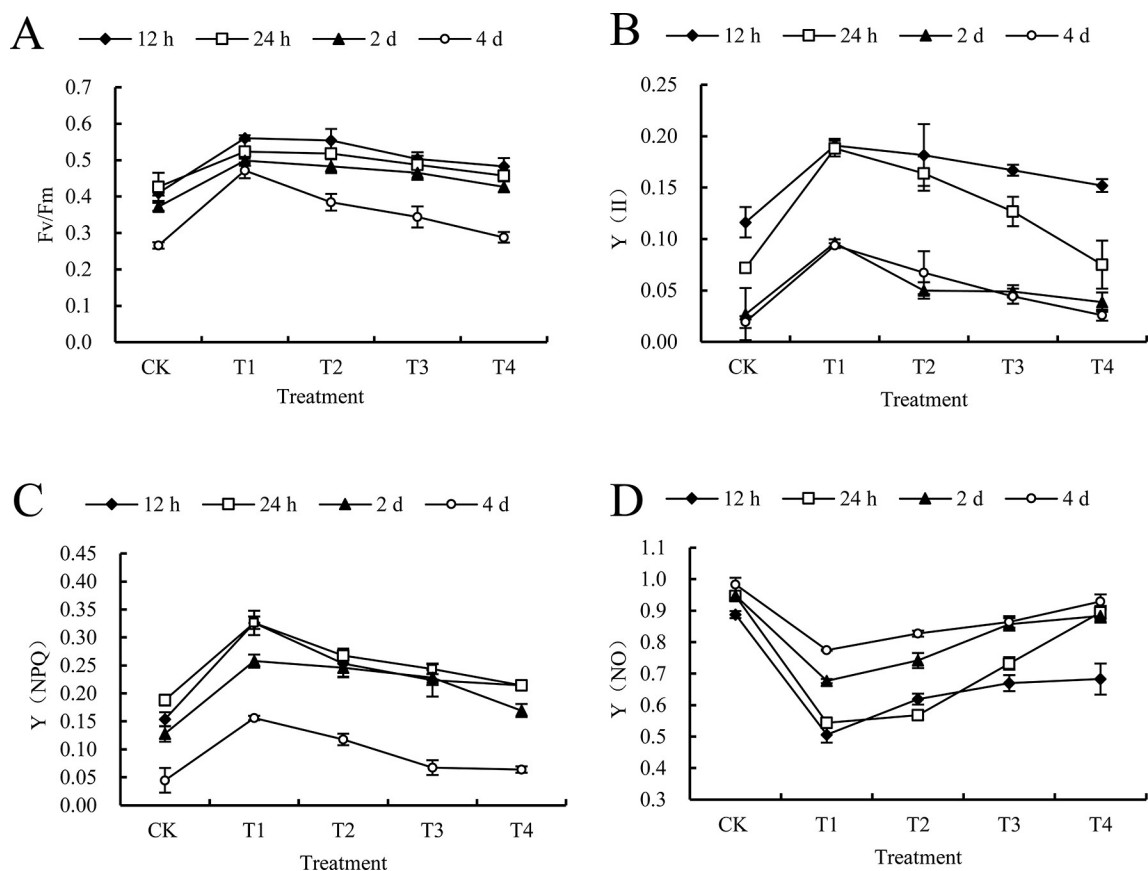

**Fig 3. Effect of 6 mm/h rain intensity on chlorophyll fluorescence parameters of *C. album* treated by mesotrione.** CK and T1, T2, T3, T4 on the x-axis represent no rain and different rainfall interval times (0.5 h, 1 h, 2 h, and 4 h) respectively; y-axis represents chlorophyll fluorescence parameters. Diamonds, squares, triangles and circles represent different times after treatment (12 h, 24 h, 2 d and 4 d, respectively). A) the maximum photochemical quantum efficiency (Fv/Fm); B) the actual photochemical quantum yield Y (II); C) the regulated energy dissipation quantum yield Y (NPQ); D) the non-regulated energy dissipation quantum yield Y (NO) of *C. album* treated by mesotrione. Lowercase letters in the same column indicate significant differences at the 0.05 level.

between other treatments and CK Y (NPQ); in addition, the Y (NO) index after T1 interval treatment had a certain drop of 10.38%. In contrast, there were not many differences between other treatments and CK (Fig 2C and 2D).

**3.4.2 Effect of 6 mm/h rain intensities on chlorophyll fluorescence parameters after spraying mesotrione.** Based on the situation shown in Fig 2, if the rain interval continued to extend, the indicators Fv/Fm and Y (II) would both show a downward trend. In particular, if the application exceeds 12 hours, the effect of mesotrione will be greatly reduced due to rainfall. In addition, the Fv/Fm index values are much higher than in the CK case in all treatment intervals, and the corresponding growth rates are 42.47%, 33.27%, 24.97% and 17.96%, respectively. In addition, on the 4[th] day, the condition of the patients treated with the T4 interval will be relieved, and the Fv/Fm index will no longer be significantly different from that in the case of CK. It should also be noted that compared with the case of CK, the Fv/Fm of other treatments will be greatly improved, and the corresponding increases are 82.59%, 72.07% and 62.67% respectively (Fig 3A).

Compared to the situation of CK, Y (II) of *C. album* had a increasement by 528.50% and 398.96% at 4 DAT for T1, and T2 treatments. And it is noted that it is almost the same as T3, T4 treatments and CK (Fig 3B), stating that if the copprresponding rain intensity

reached 6 mm/h, the rainfall within 1 hour could be give a great influence at the efficacy of mesotrione.

The Y (NPQ) value of *C. album* was measured at different periods, and its magnitude was as follows: 24 hours after treatment>12 hours after treatment>2 days after treatment>4 days after treatment. The photosynthetic inhibitory effect of *C. album* was the strongest at 4 DAT. Compared with 12 hours after application, because of the self-protection mechanism of *C. album*, Y (NPQ) will has a slower increasment, and then slowly decreased; Y (NO) showed that with the extension of the interval and the passage of the reaction time gradually increase. Compared with CK at 24 hours after the mesotrione, Y (NPQ) in all interval treatments decreased significantly; Y (NO) in all interval treatments was significantly lower than CK, T1 and T2 treatments would not show much differences (Fig 3C and 3D).

**3.4.3 The effect of 10 mm/h rain intensities on chlorophyll fluorescence parameters after spraying mesotrione.**   In addition, based on the display status of the graph, rainfall will increase the Fv/Fm and Y (II) values of *C. album* (Fig 3). However, this situation will be weakened by the extension of the rainfall time, and this trend will gradually flatten out. Compared with the results under CK conditions, Fv/Fm show the great increment by 25.14%~43.93% for all intervals at 12 h and 4 days after treatment. Ignoring the results from T4 treatment, others would demonstrate the better increment by 85.08% and 78.31%, 71.54%, which was much higher than CK, and T1, T2 and T3 treatment, Y (II) give a good growth from the observation, which means that the possible influence of mesotrione on *C. album* become much weaker when it is raining. If the rain intensity arrives at 10 mm/h, T1, T2 and T3 treatment had a much stronger restrictive effect mesotrione (Fig 4A and 4B). Y (NPQ) will fluctuate when it suffers from rain intensity of 2 mm/h and 6 mm/h. It might have a growth at 24 h after treatment compared with 12 h, and then slowly declined and minimized after 4 d, except for the T4 treatment. T1, T2 and T3 treatments is possible to have an impact on the mesotrione of *C. album*. Compared with CK, Y (NPQ) gives a higher increment at 422.82%, 220.58%, and 199.11% (Fig 4C).

## 4 Discussion

With the continuous acceleration of the process of agricultural modernization and the gradual transfer of rural labor to cities, chemical control has become the dominant method in the field of weed control due to its high efficiency and practicality [29,30]. Temperature, humidity, light, wind, and precipitation will affect the effectiveness of the mesotrione [31,32]. Global warming causes the rainfall increases and the rainfall mainly is the short-duration heavy rainfall in summer, heavy monsoon rains would increase the risk of crop failure within regions or globally [33,34]. The slow-moving cold front delays the occurrence of peak rainfall but enhances the rainfall intensity, the maximum hourly summer rainfall intensity has increased by about 11.2% on average [35]. The scouring effect of rain water reduces the effective components of the medicament attached to the leaf surface and reduces the herbicide effect [36].

Zhang *et al.* [37] reported that with the increase of the dose of mesotrione, the inhibitory effect on corn growth increased, mainly manifested as a significant decrease in plant height; Zhang *et al.* [22] and Wang *et al.* [23] reported spraying nitrate. After sulcotrione, the heart leaves of Amaranthus retroflexus were severely chlorosis and then the whole plant will died. The control efficiency of plant and fresh weight were high than common plants; Gao *et al.* [38] reported that the effect of mesotrione in the control of weeds increased with weed physiological metabolism intensified, the symptoms are more significant later. Spillman [39] think the rainfall would dilute or wash the medicament on weed's surface, and then the reduced efficacy couldn't have a thorough effect on the weed. In this study, on condition that the rain intensity

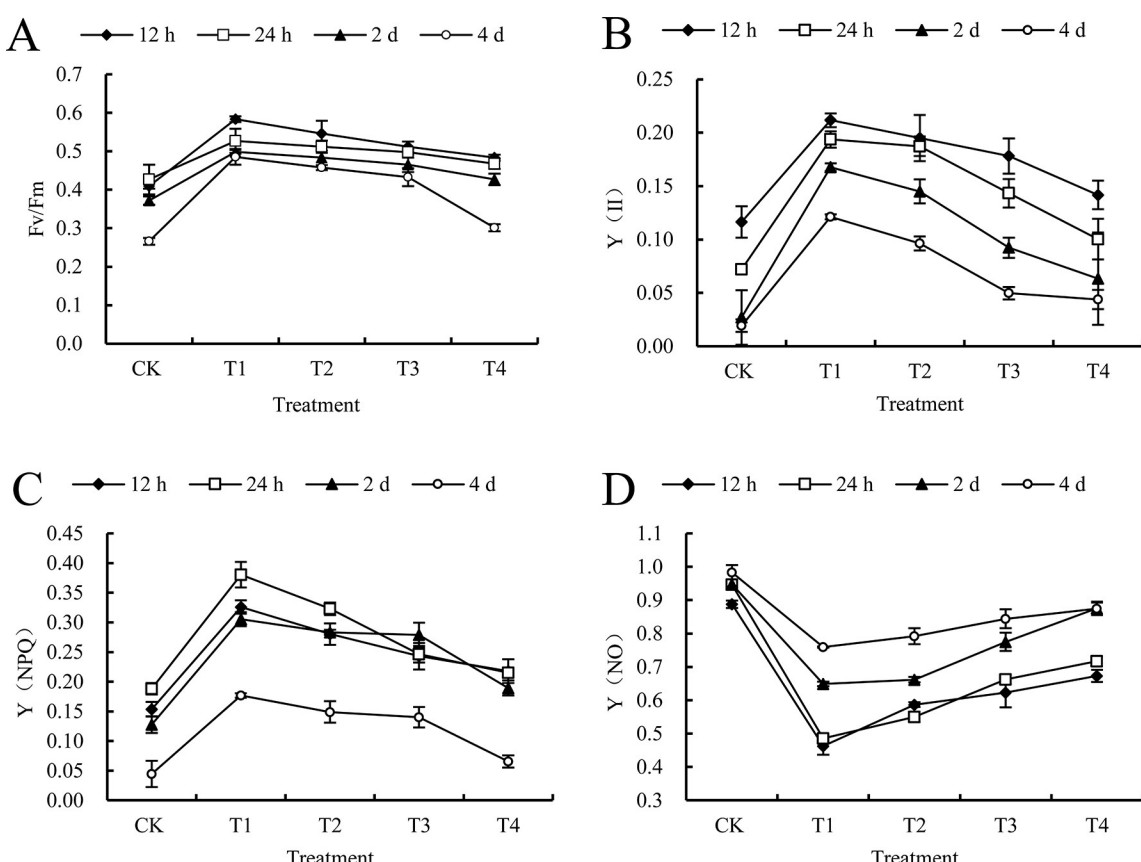

**Fig 4. Effect of 10 mm/h rain intensity on chlorophyll fluorescence parameters of *C. album* treated by mesotrione.** CK and T1, T2, T3, T4 on the x-axis represent no rain and different rainfall interval times (0.5 h, 1 h, 2 h, and 4 h) respectively; y-axis represents chlorophyll fluorescence parameters. Diamonds, squares, triangles and circles represent different times after treatment (12 h, 24 h, 2 d and 4 d, respectively). A) the maximum photochemical quantum efficiency (Fv/Fm); B) the actual photochemical quantum yield Y (II); C) the regulated energy dissipation quantum yield Y(NPQ); D) the non-regulated energy dissipation quantum yield Y (NO) of *C. album* treated by mesotrione. Lowercase letters in the same column indicate significant differences at the 0.05 level.

was 2 mm/h and the rainfall interval was T1, the plant height inhibition rate of *C. album* was still close to that of no rainfall after application, indicating that a light rain after application would not affect the efficacy of mesotrione. When the rain intensity is 6 mm/h, under the treatment of 2 h interval, with the extension of the rain interval time, the control effect of mesotrione on *C. album* gradually increases, and the control effect of plant and fresh weight is still above 90%, it only decreased by 0.4%-2.47% compared with CK, the efficacy of mesotrione was not significantly affected.

The mechanism of action of herbicides is mainly to cause weed death by interfering or inhibiting the physiological metabolism of plants, including photosynthesis, fatty acid synthesis, pigment synthesis, amino acid metabolism [40]. HPPD is the only target enzyme for mesotrione plants, and carotenoids are synthesized products of the target enzyme (HPPD) after the catalytic reaction. Therefore, the changes in the mass fraction of photosynthetic pigments in the leaves can reflect the HPPD enzyme in *C. album*. The level of activity is used as an important indicator of the weed control efficiency of mesotrione on weed control [41]. Rainfall not only straightly affected the efficacy of herbicide by changing the absorbing and conduction capability inside the plants, but also by changing the growth and physiological property indirectly. Zhang *et al*. [42] found that the carotenoid, chlorophyll a and chlorophyll b content in

the leaves of *C. album* gradually decreased with the time of the herbicide reaction after spraying mesotrione. Zhang *et al*. [43] thinks that the erosion resistance is better with the increase of retention on leaves. In this experiment, the control effect decreased after spraying mesotrione, may be due to rainfall washing the pesticide retention on the plant surface, which increased the photosynthetic pigment content of *C. album* leaves, but as the reaction time increased, the content of photosynthetic pigments decreased step by step. However, when the rainfall intensity increased, the content of carotenoids in leaves of *C. album* was significantly higher than that of the control when the rainfall intensity was 6 mm/h and the interval was within 1 h. If the rainfall intensity continued to add to 10 mm/h, the carotenoid content of leaves of *C. album* was still significantly increased after 2 h interval times, significant difference disappeared under the 4 h rainfall disposal. It is consistent with the conclusion which was the rainfall after 4 h didn't have effect on the efficacy of the new-style herbiside—Y11049 [44].

Chlorophyll fluorescence is a means of rapidly identifying injury to leaves in the absence of visible symptoms. This is a detailed analysis of causes of change in photosynthetic capacity. It has been widely used in laboratory studies in understanding the mechanism of photosynthesis and the mechanisms by which a range of external factors alter photosynthetic capacity. The measurement of chlorophyll fluorescence also has considerable potential for use in the field situation [45]. Fv/Fm represents the strength of photoinhibition after the plant is stressed, reflecting the maximum light energy conversion efficiency in the photosynthetic reaction center, and Y (II) refers to the actual light energy conversion efficiency of the plant photosynthesis process [46]. Y (NPQ) is one of the self-protection indicators for plants to convert excess light energy into heat and the degree of light damage to plants is represented by Y (NO) [47,48]. Yang *et al*. [49] found that 2,4-D treatment significantly reduced the Fv/Fm and Y (II) of Jinfen No.107 and Jingu No.29 millet seedling leaves. Guo *et al*. [50] reported that after spraying chipton, the Fv/Fm of millet has an increasing inhibitory effect as the increased of the spraying concentration. We found that Fv/Fm, Y (II), Y (NPQ) of *C. album* leaf gradually decreased with the extension of the interval, Y (NO) showed an upward trend conversely. We had tested the Y (NPQ) of the *C. album* leaf after spraying mesotrione for 24 h, it was higher than that of 12 h, indicating that the *C. album* can relieve the caused by mesotrione to a certain extent by increasing heat dissipation after being slightly stressed. Photodamage, the photochemical reaction efficiency continues to decline due to the serious herbicide stress, and it shows a downward trend as the rain interval increases. Therefore, the efficacy of mesotrione has weakened, which is mainly due to the increase in leaf photosynthetic pigment content, and the increase of photosynthetic capacity of photosystem II, which affects the control effect of mesotrione and the growth of *C. album* and photosynthesis. *C. album*'s growth difference under different disposal is directly connected with the absorption and conduction of mesotrione, it needs further discussion about the loss of the efficacy and the residual on the surface of leaves.

When the rainfall intensity was 6 mm/h within 1 h after mesotrione spraying, or the rainfall intensity was 10 mm/h within 2 h, the loss of plant control effect was above 11%. Oils adjuvants could reduce the surface tension and contact angle of the herbicide liquid, then the herbicide would have better wettability, adhesiveness, penetrating quality and resistance to rain washing capability and ultimately the efficacy would be increased [51]. Tao *et.al* [52] researched that adjuvants could increase the prevention and control effect if the rainfall appears in 4 h after mesotrione was applied. Terrible environment would reduce herbicide's efficacy, but effective ingredient in the adjuvants could increase herbicide's prevention and control effect by counteracting the influence. And how to choose adjuvants needs to be further studied.

## 5 Conclusion

In conclusion, rainfall reduced the control effect of mesotrione on *C. album*, one of the reasons was the increase of photosynthetic pigment content and the enhancement of photosystem II activity. The 2 mm/h not significantly effected the control effect of mesotrione; In the circumstance of 6 mm/h of the rainfall intensity, the control effect of mesotrione was significantly reduced within 0.5 h and 1 h, the fresh weight control efficacy decreased by 16.90% and 7.14%, respectively; and in the 10 mm/h of the rainfall intensity, the plant control efficacy was reduced by 28.48% and 18.42%, respectively, even the interval of rainfall was 2 h, it still showed a reduction rate of 14.78 percent.

## Supporting information

**S1 Table. The raw data.** The table matched to Table 4.
(XLS)

**S2 Table. The raw data.** The table matched to Fig 1.
(XLSX)

**S3 Table. The raw data.** The table matched to Table 5.
(XLSX)

**S4 Table. The raw data.** The table matched to Fig 2.
(XLSX)

**S5 Table. The raw data.** The table matched to Fig 3.
(XLSX)

**S6 Table. The raw data.** The table matched to Fig 4.
(XLSX)

## Acknowledgments

We would express sincere appreciation to two anonymous reviewers for their thoughtful and valuable comments, which helped us to improve this manuscript. Thanks are also due to Professor X.Y. Yuan of Shanxi Agricultural University for suggestions on the manuscript.

## Author Contributions

**Conceptualization:** Xie Song, Xiangyang Yuan.

**Funding acquisition:** Xiangyang Yuan.

**Investigation:** Xiaoxin Shi.

**Resources:** Shuai Guo, Yanfen Li.

**Writing – original draft:** Mengmeng Sun, Meijun Guo.

**Writing – review & editing:** Meijun Guo, Shuqi Dong.

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
