## [Decision Letter · Decision Letter 0]

30 Jun 2021

PONE-D-21-13877

Effects of mesotrione on the control efficiency and chlorophyll fluorescence parameters of Chenopodium under simulated rainfall condations

PLOS ONE

Dear Dr. Guo,

Thank you for submitting your manuscript to PLOS ONE. After careful consideration, we feel that it has merit but does not fully meet PLOS ONE’s publication criteria as it currently stands. Therefore, we invite you to submit a revised version of the manuscript that addresses the points raised during the review process.

Please find the reviewers' comments and address them point-by-point.

We look forward to receiving your revised manuscript.

Kind regards,

Debalin Sarangi, Ph.D.

Academic Editor

PLOS ONE

Journal Requirements:

3. Please include tables in Manuscript

Additional Editor Comments (if provided):

Please find the reviewers' comments and address them point-by-point.

Reviewers' comments:

Reviewer's Responses to Questions

**Comments to the Author**

1. Is the manuscript technically sound, and do the data support the conclusions?

Reviewer #1: Yes

Reviewer #2: Yes

2. Has the statistical analysis been performed appropriately and rigorously? 

Reviewer #1: Yes

Reviewer #2: Yes

3. Have the authors made all data underlying the findings in their manuscript fully available?

Reviewer #1: Yes

Reviewer #2: Yes

4. Is the manuscript presented in an intelligible fashion and written in standard English?

Reviewer #1: Yes

Reviewer #2: Yes

5. Review Comments to the Author

Reviewer #1: Title: Please correct spelling of “conditions”

I am not an expert of weed science. I believe that the manuscript was sent to me based on my experience with rainfall/irrigation science and management. Thus, I am going to limit my suggestions/comments to this aspect of the manuscript only.

Line 79: This is my main concern. The authors mentioned the rainfall intensities and intervals investigated without any context of how do these treatments hold relevance in the local climate systems.

It would be great if authors could include some justification for their choice of rainfall treatments. Ideally, I would like to see some analysis of hourly rainfall data in the last decade or a normal (30 year) period at the experimental site. To see, what was the frequency of occurrence of a particular intensity and interval at the site. This way, we will know that the experimental site is usually subject to what kind of precipitation events. The findings of the research can then be interpretable for local stakeholders.

Also, then the authors would be able to better ascertain that the application of mesotrione does not cause any efficacy loss at the experimental sites, given that in the historical rainfall data, we see that the most popular combination of intensity/interval is which one of the treatments.

It is fine if the authors chose to do this analysis only for the agricultural growing season, when weed control is relevant.

I will leave it on to the other weed science experts to peer-review this paper from that standpoint.

Reviewer #2: • Line no. 14 space

• Kindly mention control efficiency reduced how much percentage?

• Line no. 37 rewrite the sentence

• In introduction: Give at-least one example how much yield loss? What is the importance of Chenopodium weed, its biology, and morphological behavior could better explain the results.

• Mention its chemical content and concentration

• Line no. 73 materials and methods the number may be given 2

• How you controlled the rainfall intensity, duration under field condition. Also drift effect is common under field condition.

• The experiment was conducted for one season only (4 months), whereas repeated results can explain the cause and effects better.

• Line 86: manual or power operated sprayer

• Line 92: How artificial spraying mimicked the natural rain, what is the height of spraying on the weed. If you have picture just present for clear understanding

• Only chenopodium plans were measured? What about remaining weed species?

• What is the plot size?

• Study conducted in open field or green house conditions?

• Line 136: weed may be removed

• Line no. 137: mention disease like

• Line no. 203: 4th

• Results are clearly explained but, Herbicide effect was studied on weed under sole weed cultivation.

• But the response of herbicide would be different in real (mixed) condition with crop plants). Hence combined plant stand would explain better.

• No data on chenopodium leaf area, since LA is the most important factor decides the efficacy of herbicide.

• What is the retention of herbicide concentration on weed leaves?

• How much of applied herbicide was lost?

• Discussion is fine with the supporting literature. However, your results could have discussed more, for example time of rain interval, all rain intensities and interaction effect, what extent it affected.

• Discussion section is lacking the supporting data on herbicide retention on leaves, leaching loss, biomass reduction.

6. PLOS authors have the option to publish the peer review history of their article (what does this mean?). If published, this will include your full peer review and any attached files.

Reviewer #1: No

Reviewer #2: **Yes: **Hanamant M. Halli

---

## [Author Response · Author response to Decision Letter 0]

29 Aug 2021

Thank you for your letter and for the reviewers’ comments concerning our manuscript . Those precious comments are all valuable and very helpful for revising and improving our paper, as well as the important guiding significance to our researches. We have studied comments carefully and have made correction which we hope meet with approval.

---

## [Decision Letter · Decision Letter 1]

21 Feb 2022

PONE-D-21-13877R1Effects of mesotrione on the control efficiency and chlorophyll fluorescence parameters of Chenopodium album under simulated rainfall conditionsPLOS ONE

Dear Dr. Guo,

Thank you for submitting your manuscript to PLOS ONE. After careful consideration, we feel that it has merit but does not fully meet PLOS ONE’s publication criteria as it currently stands. Therefore, we invite you to submit a revised version of the manuscript that addresses the points raised during the review process.

 Please see the revision requests from the staff editors from PLOS ONE in the **Journal Requirements **section of this letter.

We look forward to receiving your revised manuscript.

Kind regards,

Bruno Jesus, Ph.D

Academic Editor

PLOS ONE

**Journal Requirements:**

PLOS ONE does not copyedit manuscripts. Please see the attached PDF with minor copyediting corrections suggested from Reviewer 1. Before PLOS can consider an accept decision for this manuscript please consider the copyediting suggestions from this reviewer in the manuscript, and proofread the manuscript.

Additional Editor Comments (if provided):

Reviewers' comments:

Reviewer's Responses to Questions

**Comments to the Author**

1. If the authors have adequately addressed your comments raised in a previous round of review and you feel that this manuscript is now acceptable for publication, you may indicate that here to bypass the “Comments to the Author” section, enter your conflict of interest statement in the “Confidential to Editor” section, and submit your "Accept" recommendation.

Reviewer #1: (No Response)

Reviewer #2: All comments have been addressed

2. Is the manuscript technically sound, and do the data support the conclusions?

Reviewer #1: Partly

Reviewer #2: Yes

3. Has the statistical analysis been performed appropriately and rigorously? 

Reviewer #1: Yes

Reviewer #2: Yes

4. Have the authors made all data underlying the findings in their manuscript fully available?

Reviewer #1: Yes

Reviewer #2: Yes

5. Is the manuscript presented in an intelligible fashion and written in standard English?

Reviewer #1: Yes

Reviewer #2: Yes

6. Review Comments to the Author

Reviewer #1: I am not sure if this was an error, but the authors have not responded to any of my comments in their response letter. I do not see my comments in there.

Reviewer #2: I appreciate your efforts in addressing the comments, Kindly address few more suggestions marked in the text especially in the materials and methods.

7. PLOS authors have the option to publish the peer review history of their article (what does this mean?). If published, this will include your full peer review and any attached files.

Reviewer #1: No

Reviewer #2: **Yes: **Hanamant M. Halli

---

## [Author Response · Author response to Decision Letter 1]

28 Mar 2022

Dear Editors and Reviewers:

 Thank you very much for your careful review on our manuscript and giving us the opportunity to resubmit our manuscript. Here we are submitting our manuscript that we have revised extensively according to your and reviewers` instructions and comments. We hope you agree that our revision has satisfactorily improved our manuscript for publication in PLOS ONE. 

Here are our responses and corrections in the paper:

1. Reviewers' comments: Line no. 30、36, add space

 Responses to Questions: Thank you for pointing this out, we have changed it.

2. Reviewers' comments: Line no. 64-66, Make the botonical name italic

Responses to Questions: We are very sorry for our negligence, we have made correction according to the Reviewer’s comments.

3. Reviewers' comments: Line no. 112——Table 1, check the min and max. temperatures

Responses to Questions: Our meteorological data was from the Taigu County Meteorological Bureau. To check the min and max temperature, we applied it to Jinzhong Meteorological Bureau again for data comparison, and the result is consistent. (Note: Taigu is a county of Jinzhong City)

4. Reviewers' comments: Line no. 114——Table 2, add space

Responses to Questions: Thank you for pointing this out, we have changed it.

5. Reviewers' comments: Line no. 118, CRD for field condition? Plz. check its CRD or randomized block design.

Responses to Questions: We are very sorry for our careless in the manuscript, we have corrected CRD to randomized block design in our revised manuscript.

6. Reviewers' comments: Line no. 120, mention the recommendation in terms of active ingredient (a.i.). The given dose may vary.

Responses to Questions: As for the reviewer’s concern, we checked the label of mesotrione again. We have changed line 120-122 as followed: “Mesotrione (10%, OD) was provided by Shandong Shengbanglvye Chemical Limited Company, the recommended effective dose given by the manufacturer was 150 g ai ha-1. The application was performed with a laboratory pot sprayer to deliver 450 L ha-1. ”

7. Reviewers' comments: Line no. 128-129, 1 L rainfall per 1 m2 = 1 mm Likewise, 1 L rainfall per 0.5 m2 = 2mm Why half an hour is mentioned here? clarify is is half an hour or half meter square.

Responses to Questions: I'm sorry I didn't write it clearly enough, “1 L rainfall per 1 m2 = 1 mm”, this is not entirely true, because it was limited by the time——one hour. It means that we have to make sure it lasted one hour when the 1 L rainfall fell on the ground whose area is 1 m2. We simulated light rain 2 mm/h in each experimental plot featured a dimension of 2 m × 2 m, it was that 8 L/h =2mm/h. If our spraying time is half an hour, it is equal to the rainfall whose intensity is 2 mm/h when the capacity of spraying was 4 L. In order to make it more clear for the audience, we have added the experimental plot size on the line 131 of the revised manuscript.

8. Reviewers' comments: Line no. 142、154、192、252, add space

Responses to Questions: Thank you for pointing this out, we have changed it.

9. Reviewers' comments: Line no. 598——Figures, Try to include clear figures as they are blurring after zooming

Responses to Questions: Thank you very much for this excellent suggestion, we have resumitted some pictures which are more clearer.

In all, I found the reviewer’s comments are quite helpful, and I revised my paper point-by-point. Thank you and the review again for your help.

---

## [Editor Report · Decision Letter 2]

13 Apr 2022

Effects of mesotrione on the control efficiency and chlorophyll fluorescence parameters of Chenopodium album under simulated rainfall conditions

PONE-D-21-13877R2

Dear Dr. Guo,

We’re pleased to inform you that your manuscript has been judged scientifically suitable for publication and will be formally accepted for publication once it meets all outstanding technical requirements.

Kind regards,

Bruno Jesus, Ph.D

Academic Editor

PLOS ONE
---

## [Editor Report · Acceptance letter]

25 May 2022

PONE-D-21-13877R2 

Effects of mesotrione on the control efficiency and chlorophyll fluorescence parameters of *Chenopodium album* under simulated rainfall conditions 

Dear Dr. Guo:

I'm pleased to inform you that your manuscript has been deemed suitable for publication in PLOS ONE. Congratulations! Your manuscript is now with our production department. 

Kind regards, 

on behalf of

Dr. Bruno Jesus 

Academic Editor

PLOS ONE